# Effect of different long-term fertilizer managements on soil nitrogen fixing bacteria community in a double-cropping rice paddy field of southern China

**Haiming Tang©\*, Chao Li, Lihong Shi, Xiaoping Xiao, Kaikai Cheng, Li Wen, Weiyan Li**

Hunan Soil and Fertilizer Institute, Changsha, China

\* tanghaiming66@163.com

**Data Availability Statement:** All relevant data are within the manuscript and its Supporting Information files.

## Abstract

Soil microorganism plays an important role in nitrogen (N) fixation process of paddy field, but the related information about how soil microorganism that drive N fixation process response to change of soil phy-chemical characteristics under the double-cropping rice (*Oryza sativa* L.) paddy field in southern of China is need to further study. Therefore, the impacts of 34-years different long-term fertilization system on soil N-fixing bacteria community under the double-cropping rice paddy field in southern of China were investigated by taken chain reaction-denaturing gradient gel electrophoresis (PCR-DGGE) method in this paper. The field experiment were set up four different fertilizer treatments: chemical fertilizer alone (MF), rice straw and chemical fertilizer (RF), 30% organic manure and 70% chemical fertilizer (OM), and unfertilized as a control (CK). This results showed that compared with CK treatment, the diversity index of *cbbL*R and *nifH* genes with OM and RF treatments were significantly increased ($p<0.05$), respectively. Meanwhile, the abundance of *cbbL*R gene with OM, RF and MF treatments were increased by 23.94, 12.19 and $6.70 \times 10^7$ copies g$^{-1}$ compared to CK treatment, respectively. Compared with CK treatment, the abundance of *nifH* gene with OM, RF and MF treatments were increased by 23.90, 8.82 and $5.40 \times 10^9$ copies g$^{-1}$, respectively. This results indicated that compared with CK treatment, the soil autotrophic azotobacter and nitrogenase activities with OM and RF treatments were also significantly increased ($p<0.05$), respectively. There were an obvious difference in features of soil N-fixing bacteria community between application of inorganic fertilizer and organic manure treatments. Therefore, this results demonstrated that abundance of soil N-fixing bacteria community in the double-cropping rice paddy field were increased by long-term applied with organic manure and crop residue managements.

## Introduction

Biological nitrogen (N) fixation, the mediated transformation of nitrogen gas (N$_2$) into ammonia by soil bacteria, is usually regarded as a vital process in maintaining reliable N-supply for

**Funding:** This study was supported by National Natural Science Foundation of China in the form of a grant awarded to HT (31872851), Innovative Research Groups of the Natural Science Foundation of Hunan Province in the form of a grant awarded to HT (2019JJ10003), and Hunan Agricultural Science and Technology Innovation Fund Project in the form of a grant awarded to HT (2021CX36).

**Competing interests:** The authors have declared that no competing interests exist.

rice growth in paddy field [1]. Compared with upland soil, the N accumulation in paddy soil were increased by 32.0 kg ha$^{-1}$ per year through soil biological $N_2$-fixation process [2]. This was ascribed to the promoting of soil nitrogenase (*nifH*) and $N_2$-fixing bacteria (*cbbL*R) activities under anaerobic flooded condition [3,4]. In the previous studies, these results indicated that soil bacteria with $N_2$-fixation capacity were distributed through diverse prokaryotic taxa including *Proteobacteria* (α-, β-, γ- and δ-proteobacteria), Clostridia and phototrophic Cyanobacteria [1,5,6]. The process of soil $N_2$-fixing bacteria in paddy field were remain largely unclear, therefore, it was benefit practice for improving rice production with lower N-fertilizer application through understanding the change of soil $N_2$-fixing bacterial community and enhancing soil biological $N_2$-fixation in paddy field [7].

In recent years, the results indicated that soil bacteria community structure and function were more and more explored by using 16S rRNA gene sequencing technology [8]. It have been successfully applied with investigate soil $N_2$-fixing bacteria community by using molecular technology, such as denaturing gradient gel electrophoresis (DGGE) [9], quantitative polymerase chain reaction (qPCR) and cloning [10], PCR-restriction fragment length polymorphism (RFLP), and fluorescently labelled terminal (FLP)-RFLP [11,12], and so on. These molecular technologies were provided more detail information about the $N_2$-fixing bacteria community in different environment (soil, rhizosphere of native wetland specie, and continental margin sediment) for researcher, compared to tradition culture methods [13]. These results indicated that N-fixing bacteria were existence mainly at upper soil layer (5 cm depth) and were estimated to occupy about 5% of the total soil bacterial population, as well as showed that soil N-fixing bacteria activity and community were obvious affected by different environmental factors, such as soil biogeochemical characters [12]. However, there is still limited information about effects of different long-term fertilizer managements on soil N-fixing bacterial activity and community in the double-cropping rice paddy field by using gene sequencing technology.

Rice (*Oryza sativa* L.) is one of mainly crop in Asia region, and double-cropping rice system (early rice and late rice) is the mainly cropping system in southern of China [14]. It is a beneficial practice for maintaining or improving soil quality and fertility of paddy field by applied with fertilizers (inorganic fertilizer, organic fertilizer) [15,16]. Different fertilizer managements may obvious effects on soil phy-chemical properties such as pH, soil bulk density, soil organic carbon (SOC) content [15], which in turn affecting soil N-fixation and soil microbiological characters of paddy field. We hypothesized that soil N-fixing bacterial community and diversity in the double-cropping rice paddy field were changed under taken different long-term fertilization conditions. Therefore, the 34-years long-term field experiment with different fertilizer treatments were set up in a double-cropping rice paddy field in southern of China. Hence, the aim of this research was: (1) to explore the diversity of soil N-fixing bacteria in paddy field under different long-term fertilization conditions; (2) to analyses the soil diversity of *cbbL*R and *nifH* genes and its phylogenetic with different fertilizer regime in a double-cropping rice system by using polymerase chain reaction-denaturing gradient gel electrophoresis (PCR-DGGE) method.

## Materials and methods

### Sites and cropping system

The field experiment were began in 1986 and were located in NingXiang County (28°07′ N, 112°18′ E) of Hunan Province, China. The more detail information about climatic conditions of the field experiment region, the soil phy-chemical characters at 0–20 cm layer before beginning this field experiment and cropping system were described as by Tang *et al.* (2018) [17].

## Experimental design

The field experiment were set up four fertilizer treatments: chemical fertilizer alone (MF), rice straw and chemical fertilizer (RF), 30% organic manure and 70% chemical fertilizer (OM), unfertilized as a control (CK). A randomized complete block design were applied for each fertilizer treatment distribution in the paddy field, with three replications of each fertilizer treatment, and the area of plot with each fertilizer treatment were 66.7 $m_2$ ($10.0 \times 6.67$ m). The field experiment ensured that the same total number of nitrogen (N), phosphorus pentoxide ($P_2O_5$), potassium oxide ($K_2O$) for MF, RF and OM treatments during early rice and late rice whole growth period, respectively. During early rice and late rice whole growth period, the total number of N, $P_2O_5$ and $K_2O$ for MF, RF and OM treatments were 142.5, 54.0, 63.0 kg ha$^{-1}$ and 157.5, 43.2, 81.0 kg ha$^{-1}$, respectively. More detailed information about the other fertilizer managements and filed arrangement were described as by Tang *et al.* (2018) [17].

## Soil sampling collect and soil sample preparation

Soil samples were collected from each plot in 25 August 2019, at the tillering stage of late rice. The soil samples were collected close to the rice plant at 0–20 cm layer in paddy field. Therefore, one composite soil samples consisting of twenty cores were collected from each plot, thus, three composite soil samples were taken from each fertilizer treatment at sampling time [16]. Then the fresh soil samples were placed in ice box and transported to the laboratory. The soil samples with each fertilizer treatment were divided into two parts: one part were stored at 4°C for investigate soil autotrophic azotobacter and nitrogenase activities, and the other part were stored at -80°C for conduct molecular analysis.

## Soil laboratory analysis

**Soil autotrophic azotobacter and nitrogenase activities.** Soil autotrophic azotobacter were investigated by using the following method, which described as by Li et al. (2008) [18] and Tang et al. (2021) [19]. Briefly, the number of soil autotrophic azotobacter were investigated base on the method of plate count, and the Ashby medium were used as medium, which were expressed as the number of colony per gram (g) of fresh soil (cfu g$^{-1}$).

The soil nitrogenase activity were investigated using the acetylene ($C_2H_2$) reduction method described as by Schwinghamer et al. (1980) [20]. Briefly, the fresh soil samples (7 g) were incubated in a 100 mL sterile flask with a rubber stopper. In order to obtain 1 mg C g$^{-1}$ compost, the soil samples were modified with solution containing glucose. Secondly, 10% of the air in the flask were replaced by acetylene gas and the flask were incubated in the dark at 28°C for 48 h. The ethylene ($C_2H_4$) produced of soil samples were investigated with gas chromatography by using a flame ionization detector (Trace GC UItra, Thermo-Fisher, USA). The soil nitrogenase activity were expressed as 1 nmol $C_2H_4$ per gram per hour of soil samples ($C_2H_4$ nmol/ (g·h)).

**Genomic DNA extraction and quantitative polymerase chain reaction (qPCR) investigation of the *cbbL*R and *nifH* genes.** Before extraction the DNA, in order to keep the same level of water content in different soil samples, the soil samples were freeze-dried in a freeze dryer (Songyuan, Beijing, China). Then the freeze-dried soil samples were crushed and sieved through 1 mm pore filters by using an ultra-centrifugal mill (ZM200, Retsch, Germany). DNA was extracted from the total microbial community by using 0.5 g of the freeze-dried soil samples with a FastDNA Spin Kit (MP Biomedicals, LLC, Illkirch, France), according to the manufacturer's instructions. The consent and quality of DNA were decided by using an Epoch Multi-Volume Spectrophotometer System (BioTek, USA). Then the extracted DNA were stored at -20°C condition.

The copy number of *nifH* gene in soil N-fixing bacteria were investigated by using qPCR method, which were conducted in three repetitions with an iCycler IQ5 Thermocycler (Bio-Rad, USA) by using following primer sets: PolF (5′-TGCGAYCCSAARGCBGACTC-3′) and PolR (5′-ATSGCCATCA TYTCRCCGGA-3′) [8,21,22]. Each reaction system comprise a 20 μL volume, was containing 10 μL of 2 × UltraSYBR Mixture (Cwbiotech, Beijing, China), 2 μL of DNA template, 0.4 μL (10 μM) of each primer, and the final volume were adjusted with sterile water. The qPCR reaction were analysis with an initial denaturation steps at 95˚C for 10 min, followed by 40 cycles at 95˚C for 10 s, 60˚C for 30 s, and 72˚C for 32 s. The data were retrieved at 72˚C for 10 min. The qPCR reaction were conducted with three times.

The copy number of *cbbL*R gene in soil N-fixing bacteria were performed by using qPCR method [23], which were conducted in three repetitions with an iCycler IQ5 Thermocycler (Bio-Rad, USA) by using following primer sets: *cbbL*R (5′-AAG GAY GAC GAG AAC ATC-3′) and *cbbL*RintR (5′-TGC AGS ATC ATG TCR TT-3′). The PCR reaction system of *cbbL*R gene were similar to *nifH* gene. The qPCR reaction were analysis with an initial denaturation steps at 95˚C for 15 min, followed by 40 cycles at 91˚C for 1 min, 55˚C for 1 min, and 72˚C for 2 min. The data were retrieved at 68˚C for 10 min. The quality of PCR product were decided in 1.5% agarose gel electrophoretic.

The PCR product amplified from soil samples were used for obtain a standard curve for the *nifH* gene product. The standard curve for qPCR were obtained by using tenfold serial dilution of linearized plasmids containing the cloned *nifH* gene. The range of template copies with $1.34 \times 10^5$ to $1.34 \times 10^9$ were used for produce the standard curve. A melting curve were produced at the end of the reaction to prove the specificity of amplicon. The standard curve indicated a PCR amplification efficiency of 88.5% and linearity of 0.99. The PCR product amplified from soil samples were used to produce the standard curve for the *cbbL*R gene. The standard curve for qPCR were obtained by using tenfold serial dilution of linearized plasmids containing the cloned *cbbL*R gene. The range of template copies with $3.10 \times 10^5$ to $3.10 \times 10^9$ copies were used for produce the standard curve. A melting curve were generated at the end of the reaction to prove the specificity of amplicon. The standard curve suggested a PCR amplification efficiency of 98.5% and linearity of 0.99.

**PCR and denaturing gradient gel electrophoresis (DGGE) analysis.** The primer sets used for PCR amplification were similar to those used for qPCR. A GC clamp were attached to the forward primer (CGCCCGG GGCGCGCC CCGGGCGGGGCGGG GGCACGGGGGG) to prevent complete separation of DNA strand during DGGE analysis [22]. Each PCR reaction were conducted in a 20 μL reaction mixture, were containing 1 μL of DNA template, 0.4 μL (10 μM) of each primer, 10 μL of 2 × Power Taq PCR MasterMix (Cwbiotech, Beijing, China), and the final volume were adjusted to 20 μL with sterile water. PCR amplification were investigated with a MyCycler thermal cycle (Bio-Rad, Hercules, CA, USA) by using following cycling conditions: initial denaturation steps at 94˚C for 5 min, followed by 40 cycles at 94˚C for 1 min, annealing at 58˚C for 1.5 min, and extension at 72˚C for 1.5 min, final extension at 72˚C for 10 min and cooling to 4˚C. The correct length of PCR product were detected by using electrophoresis on 1% agarose gel.

DGGE analysis were conducted by using a Dcode^TM Universal Detection System instrument (Bio-Rad, USA), according to the manufacturer's instructions. The 20 μL PCR products were loaded onto 8% polyacrylamide gel with a denaturing gradient of 30–60%. Electrophoresis were performed in 1× TAE buffer at 60˚C for 12 h at a constant voltage of 80 V (DcodeTM Universal Detection System, Bio-Rad, USA). After electrophoresis, the gels were stained with 1: 10000 DuRed (Sigma, USA) in the dark for 30 min and then photographed with UV light by using Gel Doc XR System (Bio-Rad, Hercules, CA, USA). The results of PCR-DGGE profile of soil microbial *cbbL*R and *nifH* genes were indicated in S1 and S2 Figs.

**Cloning and sequencing.** After DGGE, the obvious bands were resection from polyacrylamide gel. These bands were grind in TE buffer (30 μL) and were stored at 4°C condition. The supernatant were used as template for PCR to sequence DNA bands by using primer sets without GC-clamp. Then, suitable PCR products were transmitted to OE Biotech Company (Shanghai, China) for gene sequencing. The sequence were assembled and compared by using BLAST via the National Center for Biotechnology Information (NCBI) (http://www.ncbi.nlm.nih.gov). Sequence analysis and operational taxonomic units (OTUs) were identified according to the method described as by Fagen et al. (2012) [24]. A neighbor-joining tree were constructed by using MEGA4 software according to the Kimura two-parameter method. A bootstrap consensus tree were deduced from 1000 replicates to represent the evolutionary history of the analyzed taxa.

## Statistical analysis

DGGE image were analyzed by using Quantity One software (version 4.6.2, Bio-Rad, USA). The similarity of community fingerprints were analyzed by using unweighted pair group method with arithmetic mean to calculated the hierarchical clusters. The diversity of soil N-fixing bacteria were calculated, and were expressed as diversity index, richness index, and evenness index [19,25].

The statistical analysis of each investigated items in the present manuscript were calculated by using SAS 9.3 software package [26]. The means of each investigated items with different fertilizer treatments were conducted by using one-way analysis of variance (ANOVA) following standard procedures at the $p < 0.05$ probability level. The results were expressed as mean and standard error.

## Results

### Soil autotrophic azotobacter and nitrogenase activities

The number of soil autotrophic azotobacter with all fertilizer treatments (MF, RF, OM and CK) were ranged from 6.25 to 17.34 $\times 10^5$ cfu g$^{-1}$. This results showed that number of soil autotrophic azotobacter with MF, OM and CK treatments were significantly lower ($p < 0.05$) than that of RF treatment. Compared with CK treatment, the number of soil autotrophic azotobacter with RF, OM and MF treatments were increased 2.77, 2.27 and 1.88 times, respectively. This results indicated that number of soil autotrophic azotobacter with MF and OM treatments were significantly higher ($p < 0.05$) than that of CK treatment (Fig 1A).

The soil nitrogenase activity with all fertilizer treatments were ranged from 2.54 to 8.13 $C_2H_4$ nmol/(g·h). This results showed that soil nitrogenase activity with MF and CK treatments were significantly lower ($p < 0.05$) than that of OM and RF treatments. Compared with CK treatment, the soil nitrogenase activity with OM, RF and MF treatments were increased 3.20, 3.09 and 1.74 times, respectively. Meanwhile, the results indicated that soil nitrogenase activity with MF treatment were significantly higher ($p < 0.05$) than that of CK treatment (Fig 1B).

### Diversity of *cbbL*R and *nifH* genes

The diversity index of *cbbL*R and *nifH* genes were obvious changed under application of different long-term fertilizer treatments condition (Table 1). Compared with CK treatment, the diversity index of *cbbL*R gene with MF treatment were significantly increased ($p < 0.05$), and the diversity index of *cbbL*R gene with MF treatment were increased by 11.38% compared to CK treatment. Compared with CK treatment, the richness of *cbbL*R gene with OM, RF and

(a)

(b)

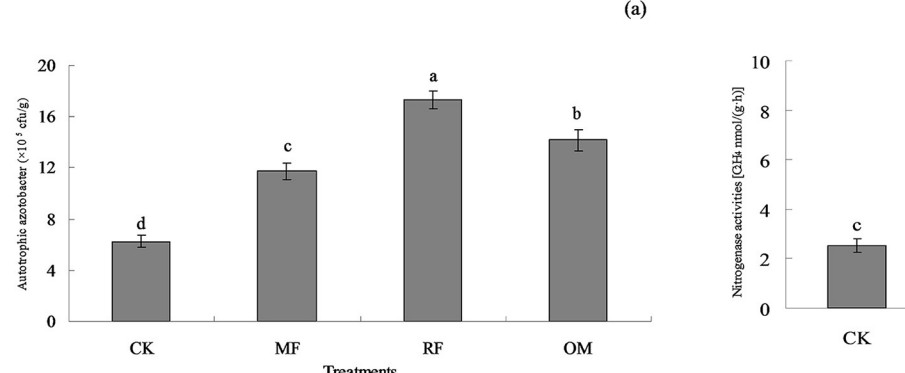
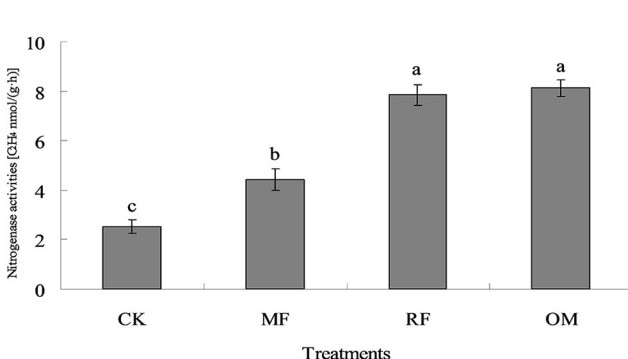

**Fig 1.** Effects of different long-term fertilizer treatments on soil autotrophic azotobacter (a) and soil nitrogenase activities (b). MF: Chemical fertilizer alone; RF: Rice straw residue and chemical fertilizer; OM: 30% organic manure and 70% chemical fertilizer; CK: Unfertilized as a control. Error bars were represented standard error of mean. Different lowercase letters were indicated significantly different at $p<0.05$ level. The same as below.

MF treatments were significantly increased ($p<0.05$). However, there were no significantly difference ($p>0.05$) in evenness of *cbbL*R gene between OM, RF, MF and CK treatments.

The results indicated that diversity index of *nifH* gene with CK treatment were significantly lower ($p<0.05$) than that of MF treatment, and this diversity index with MF treatment were increased by 11.22% compared to CK treatment. This results showed that richness index of *nifH* gene with OM, RF and MF treatments were significantly higher ($p<0.05$) than that of CK treatment. But there were no significantly difference ($p>0.05$) in evenness index of *nifH* gene between OM, RF, MF and CK treatments. That is, the diversity index of *cbbL*R and *nifH* genes were increased by long-term applied with organic manure, crop residue and chemical fertilizer managements.

## Abundance of *cbbL*R and *nifH* genes

The effects of different long-term fertilizer treatments on abundance of *cbbL*R and *nifH* genes were showed in Table 2. The abundance of *cbbL*R and *nifH* genes with MF, RF and CK treatments were significantly lower ($p<0.05$) than that of OM treatment. The results indicated that abundance of *cbbL*R and *nifH* genes with CK treatment were significantly lower ($p<0.05$) than that of MF and RF treatments, and the sequence of abundance of *cbbL*R and *nifH* genes with different fertilizer treatments were showed as OM>RF>MF>CK.

The abundance of *cbbL*R gene with all fertilizer treatments (MF, RF, OM and CK) were ranged from 1.42 to 25.36 $\times10^{7}$ copies g$^{-1}$. Compared with CK treatment, the abundance of *cbbL*R

**Table 1. Diversity of *cbbL*R and *nifH* genes were affected by different long-term fertilizer treatments.**

| Treatments | *cbbL*R | | | *nifH* | | |
|---|---|---|---|---|---|---|
| | Diversity index | Richness index | Evenness index | Diversity index | Richness index | Evenness index |
| CK | 3.25±0.10b | 2.58±0.09b | 2.26±0.07 | 3.12±0.09b | 2.51±0.06b | 2.25±0.07 |
| MF | 3.62±0.09a | 3.22±0.07a | 2.22±0.06 | 3.47±0.10a | 2.76±0.08a | 2.24±0.06 |
| RF | 3.47±0.10ab | 3.17±0.09a | 2.29±0.07 | 3.43±0.10ab | 2.70±0.08a | 2.28±0.07 |
| OM | 3.55±0.10ab | 3.11±0.09a | 2.31±0.07n.s. | 3.35±0.10ab | 2.65±0.07a | 2.32±0.07n.s. |

MF: Chemical fertilizer alone; RF: Rice straw and chemical fertilizer; OM: 30% organic manure and 70% chemical fertilizer; CK: Unfertilized as a control.

Values were presented as mean ± standard error.

n.s. means not significantly differences. Different lowercase letters in the same column were indicated significantly difference at $p<0.05$ level.

**Table 2. Abundance of *cbbL*R and *nifH* genes were affected by different long-term fertilizer treatments.**

| Gene | Treatments | | | |
|---|---|---|---|---|
| | **MF** | **RF** | **OM** | **CK** |
| *cbbL*R (×10⁷ copies g⁻¹) | 8.12±0.43c | 13.61±0.39b | 25.36±0.23a | 1.42±0.04d |
| *nifH* (×10⁹ copies g⁻¹) | 7.85±0.49c | 11.27±0.33b | 26.35±0.22a | 2.45±0.05d |

Different lowercase letters in the same line were indicated significantly difference at $p<0.05$ level.

gene with OM, RF and MF treatments were increased 17.86, 9.58 and 5.72 times, respectively. The results indicated that abundance of *nifH* gene with all fertilizer treatments were ranged from 2.45 to 26.35 ×10⁹ copies g⁻¹. Compared with CK treatment, the abundance of *nifH* gene with OM, RF and MF treatments were increased 10.76, 4.60 and 3.20 times, respectively.

## Cluster analysis of *cbbL*R and *nifH* genes

The soil community structure of *cbbL*R and *nifH* genes with all fertilizer treatments were investigated by using cluster analysis (Fig 2A and 2B). The results indicated that application of inorganic fertilizer and organic manure managements had an obvious effects on soil community structure of *cbbL*R gene, which showed that there had obvious differences in soil community structure of *cbbL*R gene between RF, CK treatments and OM, MF treatments (Fig 2A). The varied of soil community structure of *nifH* gene with all fertilizer treatments were similar to soil community structure of *cbbL*R gene (Fig 2B). These results demonstrated that fertilizer treatment were the main factor affecting soil community structure of *cbbL*R and *nifH* genes. There had three major clusters of *cbbL*R and *nifH* genes with all fertilizer treatments, OM and MF treatments were aggregated into one cluster, which indicated that there had high similarity in soil community structure of *cbbL*R and *nifH* genes between OM and MF treatments.

## Community structure of *cbbL*R and *nifH* genes

The change of soil community structure of *cbbL*R and *nifH* genes with all fertilizer treatments were investigated by using neighbor-joining phylogenetic (Figs 3 and 4), according to taxonomic affiliations of the sequences obtained in all soil samples. This results demonstrated that main dominant group of soil community structure of *cbbL*R genes were geared to *Azospira*, *Betaproteobacteria*, *Ideonella*, and *Pseudoacidovorax*. Meanwhile, this results showed that most

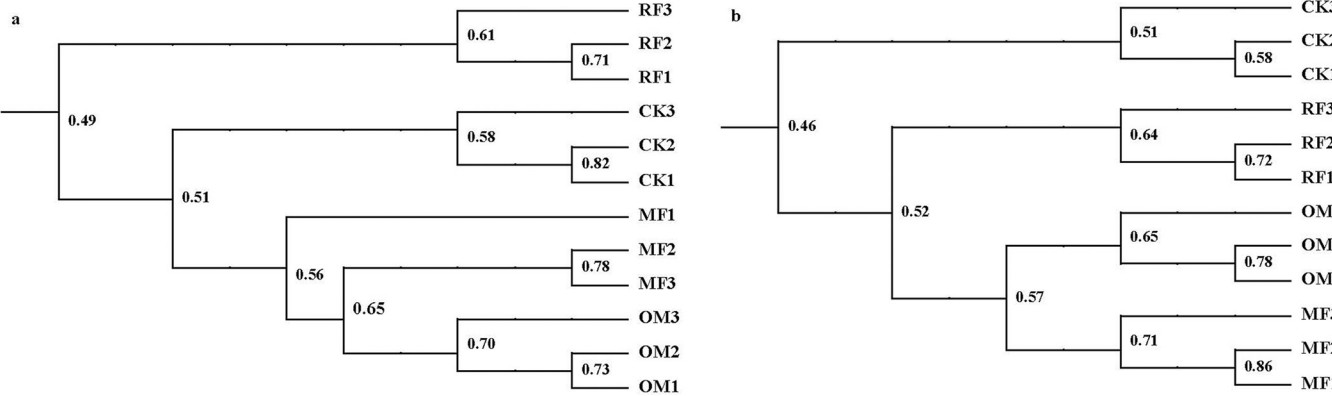

**Fig 2.** Similarity dendrograms (UPGMA, Dice coefficient of similarity) analysis of *cbbL*R gene (a) and *nifH* gene (b) with different long-term fertilizer treatments.

**Fig 3. Phylogenetic tree of *cbbL*R gene in soil samples by using all OTUs identified with different long-term fertilizer treatments.**

of *cbbL*R genes were belong to the cluster 1 lineage, and some *cbbL*R genes were belong to the cluster 2 lineage, cluster 3 lineage and cluster 4 lineage in all soil samples (Fig 3). In the first cluster, Band 9, 10, 11 and Band 13 were clustered with soil N-fixing bacteria HM565436.3, Band 5 were clustered with soil bacteria HQ335728.2, Band 14 were clustered with *Ideolla decloratans* strain EU542647.1. In the second cluster, Band 4, 6 and Band 7 were similar to *Pseudoacidovorax*. Band 1, 5, Band 8 and *Azospira* were affiliated with the third cluster, Band 12 and Band 15 were significantly differences from the other sequences.

The phylogenetic analysis were conducted based on the sequence of *nifH* gene and the most similar sequence in GenBank. This results demonstrated that most of *nifH* genes were belong to the cluster 1, Band 2, 3 and Band 6 were belong to cluster 2, while Band 1, 4 and Band 5 were belong to cluster 3. In the database of GenBank, the culture sequences of soil microorganism gene were affiliated with Band 3 and Band 4 could be resumed, the sequence of *myzf* gene were similar to the other sequences of culture soil microorganism gene, and only some uncultured sequences of soil microorganism gene were affiliated with them.

## Discussion

### Effects of fertilizer regime on soil autotrophic azotobacter and nitrogenase activities

The results of the present study suggested that number of soil autotrophic azotobacters were significantly increased under long-term application of crop residue and organic manure condition, in agreement with Yuan et al. (2011) [27]. The reason is maybe that associated with the number of soil autotrophic azotobacters were stimulated under application of fertilizers, which provide more energy substrates for soil autotrophic azotobacters growth and multiplying. In the different fertilizer treatments, the number of soil autotrophic azotobacters with OM and RF treatments were higher than that of MF and CK treatments, which suggests that soil organic carbon (SOC) content were mainly factor for promoting soil autotrophic azotobacters growth and multiplying [15]. On the other hand, the total input of organic carbon (C) in organic manure and crop residue were significantly enhanced, and the components of organic C were also changed [28]. In the present study, the results showed that number of soil autotrophic azotobacters with MF treatment were lower than that of OM and RF treatments, which suggests that soil autotrophic azotobacters in paddy field were limited under long-term application of chemical fertilizer alone condition.

The soil nitrogenase activity is a vital indicator for estimating the capacity of soil biological nitrogen (N) fixation [8]. The results of the present study indicated that soil nitrogenase activity were significantly promoted under long-term application of organic manure and crop residue condition, higher residue organic matter in OM, RF and MF treatments paddy soil than that unfertilized paddy soil maybe lead to higher nitrogenase activity in OM, RF and MF treatments paddy soil than unfertilized paddy soil [29]. The reason may be attribute to that soil N-fixing microbe activity were stimulated under application of fertilizer condition, which provides more energy substrates for soil nitrogenase growth and multiplying. Meanwhile, regarding the different fertilizer treatments, this study demonstrated that soil nitrogenase activity with OM treatment were higher than that of MF, RF and CK treatments, which suggested that SOC content was a vital factor for promoting the growth of soil nitrogenase. The soil nitrogenase activity with OM treatment were higher than that of RF treatment, maybe because crop residue treatment stimulate the N-fixation activity of soil microbe, but the restrain of N content on soil nitrogenase activity were higher than that of crop residue effects, then the soil nitrogenase activity were reduced. These results were also demonstrated that the series of soil N-fixing bacteria and continuous consume of organic manure in paddy field. However, the soil nitrogenase activity were limited by application of chemical fertilizers compared to application of organic manure and crop residue managements, which suggested that soil nitrogenase activity decreased with lower SOC content in paddy field caused by long-term application of chemical fertilization. In the present study, soil nitrogenase activity were enhanced with application of organic manure and crop residue managements in agreement with the previous studies [6,30].

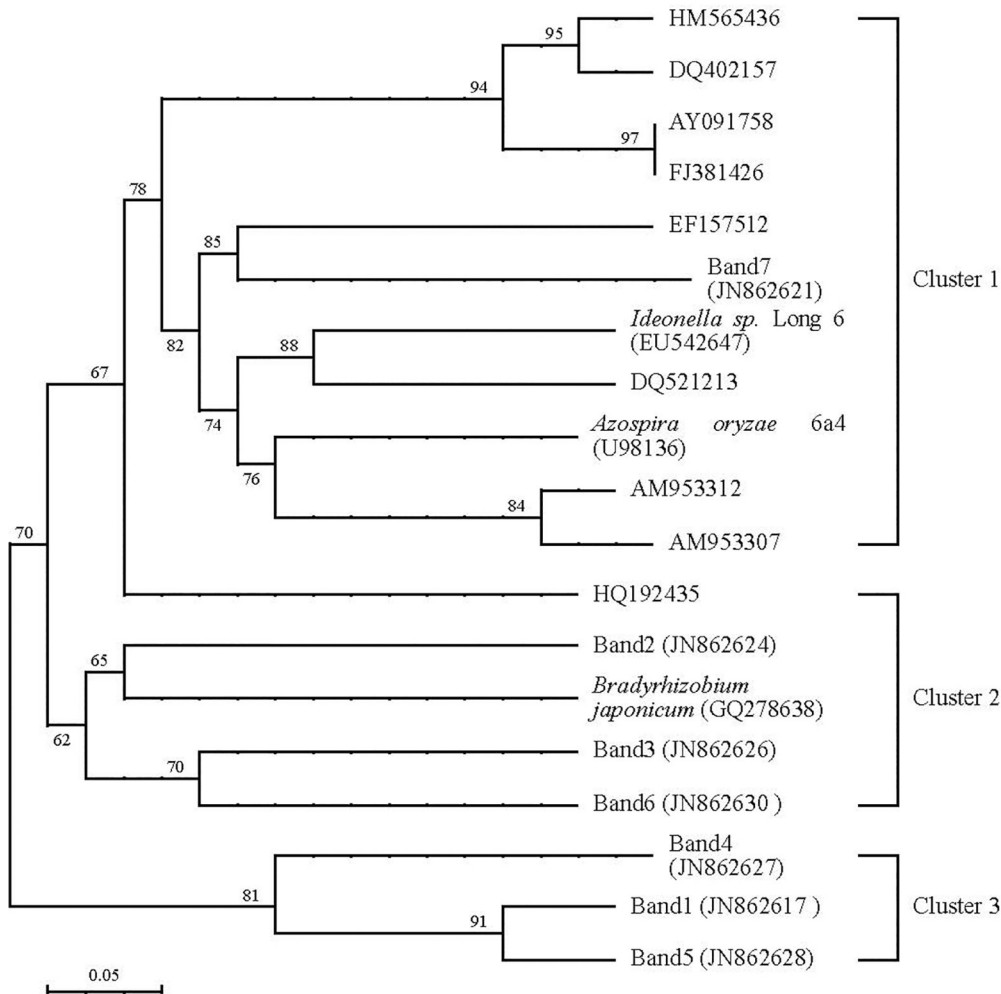

**Fig 4. Phylogenetic tree of *nifH* gene in soil samples by using all OTUs identified with different long-term fertilizer treatments.**

### Effects of fertilizer regime on soil community structure of *cbbL*R and *nifH* genes

In the previous studies, the results suggested that differences in soil phy-chemical characteristics were mainly factor affecting the soil N-fixing microorganism community structure in paddy field [31]. In this study, the results showed that community structure of soil N-fixing bacteria were more similar to in different fertilizer treatments, which had closer effects. Soil autotrophic azotobacter and nitrogenase activities were obvious changed under different soil ecological environmental properties condition. In the present study, the different fertilizer treatments were one of the important differences in the all soil samples. This results suggested that fertilizer managements may have been the vital factor influencing on the portion of soil N-fixing bacteria community structure. Long-term application of organic manure and crop residue managements (OM and RF treatments) had lead to higher diversity index of *cbbL*R and *nifH* genes in paddy soil, while long-term application of chemical fertilizers (MF treatment) had lead to lower diversity index of *cbbL*R and *nifH* genes in paddy soil. Limmer and Drake (1996) [32] results proved that activity and distribution of soil N-fixation bacteria were

mainly influenced on by soil C and N contents. In our previous study, the results showed that soil C and N contents were the highest with OM and RF treatments [15], and the highest number of clones and diversity of soil *cbbL*R and *nifH* genes. Contrary, the lowest soil C and N contents with MF and CK treatments, as well as had the lowest number of clones and diversity of soil *cbbL*R and *nifH* genes, respectively. Therefore, the soil C and N contents in paddy field may be the most important factor affecting on community structure of soil N-fixing bacterial.

In the previous studies, these results indicated that more molecular diversity and phylogenetic analysis of *cbbL*R and *nifH* genes in different geochemical environments were found by using polymerase chain reaction (PCR) method, which were provided more abundant of uncultivated *cbbL*R and *nifH* gene sequences for researcher [9,12,27]. In the present study, the results indicated that some relevant soil microorganism were affiliated with *Azospira*, *Betaproteobacteria*, *Pseudacidovorax*, *Proteobacteria* and *Ideonella*, and unidentified N-fixing bacteria were also found in the all soil samples, respectively. This results indicated that molecular diversity of soil azotobacter in paddy field were significantly increased, and Jia et al. (2020) [8] also demonstrated that it was fit for increasing rhizosphere soil biological N-fixation in paddy field. In the present study, some unidentified *cbbL*R and *nifH* gene sequences were also found. The phylogenetic tree suggested that some new soil N-fixing bacteria in the double-cropping rice paddy field were found. (i) The phylogenetic tree of *cbbL*R and *nifH* gene sequences did not indicated that there had obvious related with the soils sampled in different fertilizer treatments; however, there had obvious difference in the number of soil N-fixing bacteria among different fertilizer treatments, which suggested that soil environmental characteristics might have significantly influence on diversity and distribution of relate soil N-fixing microbial. (ii) The mainly culture clone sequences were not closely related with any identified soil N-fixing bacteria. 55% of the exhibited clone sequences were less than 72% nucleotide acid identified with known soil N-fixation bacteria in the database of GenBank, which suggested that they were distinctive and were represent new clone sequences of soil N-fixing community structure, and that majority member of soil *cbbL*R and *nifH*-containing bacteria community structure maybe uncultured. This results indicated that the new cluster of soil N-fixation bacteria looks like be higher number in the present experiment area, which suggested that clone library represent the in soil situ microbial community structure at functional cluster level. It will need to further investigate or quantify these new clusters for prove them abundance and understand them functionality in paddy soil by using real-time PCR and other molecular methods.

In this study, our results first found that community and structure of soil N-fixing bacteria with different long-term fertilizer treatments under the double-cropping rice paddy field in southern of China, thus obtaining more detail information about their genetic diversity, finding more culture clone sequences of soil N-fixation bacteria in the double-cropping rice paddy field, and also presenting a new view for fertilizer managements factors affecting these vital group of soil bacteria. For better understanding the relationship between community structure of soil microbial and functional processes involved in N cycling, it necessary further to investigate the effects of different rhizosphere soil environmental factors on community structure of soil N-fixation under long-term fertilization condition.

## Conclusion

The results of the present study indicated that number of cultivable soil N-fixing microorganism were significantly increased under long-term application of fertilizer treatments condition, with the highest number of soil N-fixing microorganism investigated under combined application of organic manure and crop residue with chemical fertilizer condition, followed by application of chemical fertilizer condition. Moreover, this results showed that diversity index and

richness index of soil *cbbL*R and *nifH* genes were enhanced under long-term combined application of organic manure and crop residue with chemical fertilizer condition. Soil nitrogenase activity were also significantly increased under long-term combined application of organic manure and crop residue with chemical fertilizer condition, but application of chemical fertilizers reduce soil nitrogenase activity in paddy field. There were expected differences in the characteristics of soil N-fixing bacteria community structure between application of inorganic fertilizer and organic manure treatments. Therefore, improving soil N content in a double-cropping rice paddy field by application of organic manure and crop residue proved to be a beneficial practice. However, future studies are needed to explore how change of soil N-fixing bacteria community structure under different long-term fertilization practice influence on ecological function of rhizosphere soil microorganism.

## Supporting information

**S1 Fig. PCR-DGGE profiles of soil microbial *cbbLR* gene.**
(TIF)

**S2 Fig. PCR-DGGE profiles of soil microbial *nifH* gene.**
(TIF)

## Acknowledgments

We acknowledge all the staff members of Hunan Ningxiang County Agricultural and Rural Bureau, and extend special thanks to Yong Li for joining this study.

## Author Contributions

**Conceptualization:** Xiaoping Xiao.

**Funding acquisition:** Haiming Tang.

**Investigation:** Chao Li, Kaikai Cheng.

**Methodology:** Xiaoping Xiao, Li Wen.

**Resources:** Lihong Shi, Li Wen.

**Software:** Lihong Shi, Kaikai Cheng, Weiyan Li.

**Supervision:** Weiyan Li.

**Writing – original draft:** Haiming Tang.

**Writing – review & editing:** Haiming Tang.

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
