## [Decision Letter · Decision Letter 0]

19 May 2021

PONE-D-20-36146

Effect of different long-term fertilizer management on nitrogen fixing bacteria community in a double-cropping paddy field of southern China

PLOS ONE

Dear Dr.Haiming Tang,

Thank you for submitting your manuscript to PLOS ONE. After careful consideration, we feel that it has merit but does not fully meet PLOS ONE’s publication criteria as it currently stands. Therefore, we invite you to submit a revised version of the manuscript that addresses the points raised during the review process.

We look forward to receiving your revised manuscript.

Kind regards,

Balasubramani Ravindran, Ph.D

Academic Editor

PLOS ONE

Journal Requirements:

2)  We suggest you thoroughly copyedit your manuscript for language usage, spelling, and grammar. If you do not know anyone who can help you do this, you may wish to consider employing a professional scientific editing service.  

3)  Thank you for stating the following in the Acknowledgments Section of your manuscript:

[This study was supported by National Natural Science Foundation of China (31872851), Innovative Research Groups of the Natural Science Foundation of Hunan Province (2019JJ10003).]

 [The author(s) received no specific funding for this work.]

4) During our internal evaluation of the manuscript, we found significant text overlap between your submission and the following previously published works.

- https://doi.org/10.1038/s41598-020-63639-8

- https://doi.org/10.1016/j.scitotenv.2020.137633

- https://doi.org/10.1111/j.1574-6968.2006.00317.x

- https://doi.org/10.1038/s41598-020-63639-8

We would like to make you aware that copying extracts from previous publications, especially outside the methods section, word-for-word is unacceptable, even for works which you authored. In addition, the reproduction of text from published reports has implications for the copyright that may apply to the publications.

Please revise the manuscript to rephrase the duplicated text, cite your sources, and provide details as to how the current manuscript advances on previous work. Please note that further consideration is dependent on the submission of a manuscript that addresses these concerns about the overlap in text with published work.

We will carefully review your manuscript upon resubmission, so please ensure that your revision is thorough

Reviewers' comments:

Reviewer's Responses to Questions

**Comments to the Author**

1. Is the manuscript technically sound, and do the data support the conclusions?

Reviewer #1: Yes

Reviewer #2: Yes

2. Has the statistical analysis been performed appropriately and rigorously? 

Reviewer #1: Yes

Reviewer #2: Yes

3. Have the authors made all data underlying the findings in their manuscript fully available?

Reviewer #1: Yes

Reviewer #2: Yes

4. Is the manuscript presented in an intelligible fashion and written in standard English?

Reviewer #1: Yes

Reviewer #2: Yes

5. Review Comments to the Author

Reviewer #1: Dear Authors, you should address my recommendations highlighted across the text, as follows.

Introduction: you should amend the text as pointed out.

Materials and methods: the amendments should be performed as suggested.

Results: the text should be corrected according to my comments. The tables should be revised following my indications relevant to the presentations of statistical outcome and the legends should be modified as recommended.

Discussion and Conclusions: the highlighted amendments should be addressed.

References: the citation formatting should be checked.

Reviewer #2: Comments to the Author

In this study, a long-term fertilizer field experiment were utilized to investigate the effects of different fertilizer management on soil nitrogen fixing bacteria community in a double-cropping rice paddy field of southern China. Therefore, the soil autotrophic azotobacter and nitrogenase activities, diversities of cbbLR and nifH gene, abundance of cbbLR and nifH gene, cluster analysis of cbbLR and nifH gene, community structure of cbbLR and nifH gene under 34-years long-term fertilizer regime in the double-cropping rice paddy field of southern China were studied in this paper. The field experiment were well designed and performed, the results were well addressed, and obtained many valuable results in this manuscript.

Therefore, the manuscript can be published after a minor revision. And there still some places need revised, such as:

1. Please ensure your paper has continuous line number.

2．The title of this manuscript need to modify as “Effect of different long-term fertilizer management on soil nitrogen fixing bacteria community in a double-cropping rice paddy field of southern China”.

3. In the “Abstract” section,

Page 2 Line 5: the “information on” should be modified to “information about”.

Page 2 Line 7: the “the 34-year” should modified to “the 34-years”, and other places were also need to revised.

Page 2 Line 7-8: the “nitrogen fixing bacteria community under double-cropping rice field in southern China” should be modified to “soil nitrogen fixing bacteria community under the double-cropping rice paddy field in southern of China”, and other places were also need to be revised.

Page 2 Line 12-13: the “The results showed that the diversity index of cbbLR gene and nifH gene were increased with RF and OM treatments,” should be modified to “The results showed that diversity index of cbbLR gene and nifH gene with RF and OM treatments were increased”, and other places were also need to revised.

4. In the “Introduction” section,

Please be sure to check that verbal tense was used when indicating the facts (usually in the present tense).

What is your innovation in the present study? You need to add more recent references and improve your introduction.

The information about hypothesize of this manuscript were need to added in the “Introduction” section.

5. In the “Materials and methods” section,

In the “Experimental design” section, some more detail information about the fertilizer treatments need to add, such as the amount of N, P2O5, K2O.

The more detail date of soil sample was need to add.

Some professional terms need to be supplied as full name at the first time in the same section, such as PCR, DGGE, and so on.

6. In the “Results” section, the information about significance needs to add.

7. In the “Discussion” section,

Page 11 Line 5: the detail name of “other fertilizer treatments” need to added.

Page 11 Line 5: “were suggestion” should modify to “were suggested”, and other places of this manuscript need to revise.

8. In the Conclusion section,

“nitrogen fixing” should modify to “soil nitrogen fixing”, “activities” should modify to “activity”, and other places of this manuscript need to revise.

9. Some more references were need to add,

In the “Introduction” section, some more references need added, such as Page 3 Line 20-22.

In the “Disscussion” section, some more references also need added, such as Page 12 Line 19-22.

10. Overall, this study provided some useful information, and it needed modification some grammatical errors throughout the MS (as presented in some examples below). The author is encouraged to seek help from a native English speaker to refine and improve the writing.

Page 2 Line 10: “rice straw residue” should modify to “rice straw”, and other places of this manuscript need to revise.

Page 3 Line 29: “practices” should modify to “practice”, and other places of this manuscript need to revise.

Page 4 Line 20: “was” should modify to “were”, and other places of this manuscript need to revise.

6. PLOS authors have the option to publish the peer review history of their article (what does this mean?). If published, this will include your full peer review and any attached files.

Reviewer #1: No

Reviewer #2: No

---

## [Author Response · Author response to Decision Letter 0]

14 Jul 2021

<Plos One >

< ID PONE-D-20-36146 R1>

Dear Editor, 

Thank you very much for your useful comments and suggestions on our manuscript. Please convey our gratitude to the reviewers who have made useful and detailed suggestions for improvement of the manuscript. As their suggestions we have revised the language and the content with red color in the manuscript. The details of the changes are listed below in point form:

Journal Requirements:

√ The style of this manuscript were revised, and the file name of this manuscript were also revised, according to Journal suggestion.

2. We suggest you thoroughly copyedit your manuscript for language usage, spelling, and grammar.

Upon resubmission, please provide the following: The name of the colleague or the details of the professional service that edited your manuscript.

√ The language usage, spelling, and grammar of this manuscript were revised, according to Journal suggestion. The related details information about colleague were as following: Tida Ge, E-mail: sjtugtd@gmail.com.

 [This study was supported by National Natural Science Foundation of China (31872851), Innovative Research Groups of the Natural Science Foundation of Hunan Province (2019JJ10003).]

 [The author(s) received no specific funding for this work.]

√ The related information about funding-related text were removed from the manuscript. And the related information about funding-related text were added in the cover letter, thank you for added these information in the online submission form on our behalf, according to Journal suggestion.

4. During our internal evaluation of the manuscript, we found significant text overlap between your submission and the following previously published works.

- https://doi.org/10.1038/s41598-020-63639-8

- https://doi.org/10.1016/j.scitotenv.2020.137633

- https://doi.org/10.1111/j.1574-6968.2006.00317.x

We would like to make you aware that copying extracts from previous publications, especially outside the methods section, word-for-word is unacceptable, even for works which you authored. In addition, the reproduction of text from published reports has implications for the copyright that may apply to the publications.

Please revise the manuscript to rephrase the duplicated text, cite your sources, and provide details as to how the current manuscript advances on previous work. Please note that further consideration is dependent on the submission of a manuscript that addresses these concerns about the overlap in text with published work.

√ The related information about duplicated text in this manuscript were revised, and the related reference were also cited, according to Journal suggestion. Such as the following references: (1) Tang HM, Xiao XP, Li C, Pan XC, Cheng KK, Li WY, et al. Microbial carbon source utilization in rice rhizosphere and nonrhizosphere soils with shortterm manure N input rate in paddy field. Sci Rep-UK. 2020; 10: 6487. (2) Zhang YG, Li DQ, Wang HM, Xiao QM, Liu XD. Molecular diversity of nitrogen-fixing bacteria from the Tibetan Plateau, China. FEMS Microbiol Lett. 2006; 260: 134–142.

Reviewer #1: 

1. Introduction: you should amend the text as pointed out.

√ In the “Introduction” section, the related places were revised, according to Reviewer suggestion.

2. Materials and methods: the amendments should be performed as suggested.

√ In the “Materials and methods” section, the related amendments were revised, according to Reviewer suggestion.

3. Results: the text should be corrected according to my comments. The tables should be revised following my indications relevant to the presentations of statistical outcome and the legends should be modified as recommended.

√ In the “Results” section, the text were revised, the relevant to the presentations of statistical outcome and the legends were also modified, according to Reviewer suggestion.

4. Discussion and Conclusions: the highlighted amendments should be addressed.

√ In the “Discussion and Conclusions” section, the related information were revised, according to Reviewer suggestion.

5. References: the citation formatting should be checked.

√ In the “References” section, the citation formatting were checked and revised, according to Reviewer suggestion.

Reviewer #2: 

1. Please ensure your paper has continuous line number.

√ This paper were modified as continuous line number, according to Reviewer suggestion.

2．The title of this manuscript need to modify as “Effect of different long-term fertilizer management on soil nitrogen fixing bacteria community in a double-cropping rice paddy field of southern China”.

√ The title of this manuscript were modified as “Effect of different long-term fertilizer management on soil nitrogen fixing bacteria community in a double-cropping rice paddy field of southern China” , according to Reviewer suggestion.

3. In the “Abstract” section,

Page 2 Line 5: the “information on” should be modified to “information about”.

Page 2 Line 7: the “the 34-year” should modified to “the 34-years”, and other places were also need to revised.

Page 2 Line 7-8: the “nitrogen fixing bacteria community under double-cropping rice field in southern China” should be modified to “soil nitrogen fixing bacteria community under the double-cropping rice paddy field in southern of China”, and other places were also need to be revised.

Page 2 Line 12-13: the “The results showed that the diversity index of cbbLR gene and nifH gene were increased with RF and OM treatments,” should be modified to “The results showed that diversity index of cbbLR gene and nifH gene with RF and OM treatments were increased”, and other places were also need to revised.

√ In the “Abstract” section,

Page 2 Line 5: the “information on” were replaced by “information about”.

Page 2 Line 7: the “the 34-year” were replaced by “the 34-years”, and other places were also revised.

Page 2 Line 7-8: the “nitrogen fixing bacteria community under double-cropping rice field in southern China” were replaced by “soil nitrogen fixing bacteria community under the double-cropping rice paddy field in southern of China”, and other places were also need to be revised.

Page 2 Line 12-13: the “The results showed that the diversity index of cbbLR gene and nifH gene were increased with RF and OM treatments,” were replaced by “The results showed that diversity index of cbbLR gene and nifH gene with RF and OM treatments were increased”, and other places were also revised, according to Reviewer suggestion.

4. In the “Introduction” section,

Please be sure to check that verbal tense was used when indicating the facts (usually in the present tense).

What is your innovation in the present study? You need to add more recent references and improve your introduction.

The information about hypothesize of this manuscript were need to added in the “Introduction” section.

√ In the “Introduction” section,

The verbal tense was usually in the present tense when indicating the facts.

The related information about the further study were added, that is, the innovation in our manuscript were present. And some more recent references were cited, such as Tang HM, Xiao XP, Li C, Pan XC, Cheng KK, Li WY, et al. Microbial carbon source utilization in rice rhizosphere and nonrhizosphere soils with short term manure N input rate in paddy field. Sci Rep-UK. 2020; 10: 6487. 

The related information about hypothesize of this manuscript were also added in the “Introduction” section.

5. In the “Materials and methods” section,

In the “Experimental design” section, some more detail information about the fertilizer treatments need to add, such as the amount of N, P2O5, K2O.

The more detail date of soil sample was need to add.

Some professional terms need to be supplied as full name at the first time in the same section, such as PCR, DGGE, and so on.

√ In the “Materials and methods” section,

In the “Experimental design” section, some more detail information about the fertilizer treatments with different fertilizer treatments were add, such as the amount of N, P2O5, K2O.

The more detail date of soil sample were also added.

Some professional terms were also supplied as full name at the first time in the same section, such as PCR, DGGE, and so on.

6. In the “Results” section, the information about significance needs to add.

√ In the “Results” section, the information about significance were added, according to Reviewer suggestion.

7. In the “Discussion” section,

Page 11 Line 5: the detail name of “other fertilizer treatments” need to added.

Page 11 Line 5: “were suggestion” should modify to “were suggested”, and other places of this manuscript need to revise.

√ In the “Discussion” section, 

Page 11 Line 5: the detail name of “other fertilizer treatments” were revised, that is, the detail name of “other fertilizer treatments” were MF and CK treatments.

Page 11 Line 5: “were suggestion” should modify to “suggests”, and other places of this manuscript need to revise.

8. In the Conclusion section, “nitrogen fixing” should modify to “soil nitrogen fixing”, “activities” should modify to “activity”, and other places of this manuscript need to revise.

√ In the “Conclusion” section, “nitrogen fixing” were modified to “soil nitrogen fixing”, “activities” were modified to “activity”, and other places of this manuscript were also revised, according to Reviewer suggestion.

9. Some more references were need to add,

In the “Introduction” section, some more references need added, such as Page 3 Line 20-22.

In the “Discussion” section, some more references also need added, such as Page 12 Line 19-22.

√ In the revised of manuscript, some more references were added, such as Page 3 Line 20-22, Page 12 Line 19-22.

10. Overall, this study provided some useful information, and it needed modification some grammatical errors throughout the MS (as presented in some examples below). The author is encouraged to seek help from a native English speaker to refine and improve the writing.

Page 2 Line 10: “rice straw residue” should modify to “rice straw”, and other places of this manuscript need to revise.

Page 3 Line 29: “practices” should modify to “practice”, and other places of this manuscript need to revise.

Page 4 Line 20: “was” should modify to “were”, and other places of this manuscript need to revise.

√ In Page 2 Line 10: “rice straw residue” were modified to “rice straw”, and other places of this manuscript were also revised, according to Reviewer suggestion.

√ In Page 3 Line 29: “practices” were modified to “practice”, and other places of this manuscript were also revised, according to Reviewer suggestion.

√ In Page 4 Line 20: “was” were modified to “were”, and other places of this manuscript need to revise, according to Reviewer suggestion.

Editor Requirements:

1. Thank you for updating your data availability statement. You note that your data are available within the Supporting Information files, but no such files have been included with your submission. At this time we ask that you please upload your minimal data set as a Supporting Information file, or to a public repository such as Figshare or Dryad. 

Please also ensure that when you upload your file you include separate captions for your supplementary files at the end of your manuscript.

√ The related information about supplementary files were added at the end of our manuscript, and the related Supporting Information file were also upload in the submission system, according to Editor suggestion.

2. We note that your submission still has substantial overlap with the previous publication "Effect of different short-term tillage management on nitrogen-fixing bacteria community in a double-cropping paddy field of southern China" (https://doi.org/10.1002/jobm.202000608). We would like to make you aware that copying extracts from previous publications, especially outside the methods section, word-for-word is unacceptable, even for works which you authored. In addition, the reproduction of text from published reports has implications for the copyright that may apply to the publications.

Please revise the manuscript to rephrase the duplicated text, cite your sources, and provide details as to how the current manuscript advances on previous work. Please note that further consideration is dependent on the submission of a manuscript that addresses these concerns about the overlap in text with published work.

√ The related information about duplicated text in this manuscript were revised, and the related reference were also cited, according to Editor suggestion. Such as the following references: Tang HM, Li C, Cheng KK, Shi LH, Wen L, Li WY, et al. Effect of different short-term tillage management on nitrogen-fixing bacteria community in a double-cropping paddy field of southern China. J Basic Microb. 2021; 61: 241–252.

Please also kindly clarify the following points:

1. Did the authors present any new data in this submission that were not previously presented in the published article?

√ There was new data in this submission that were not previously presented in the published article.

2. Did the authors perform any additional experiments or collect any additional data that were not a part of the study from the published article?

√ There was new data in this submission, and there were not perform any additional experiments or collect any additional data that were not a part of the study from the published article.

The revised manuscript has been submitted to your journal. Once again, thank you for your help and support during the process of the improvement of the manuscript and we look forward to your positive response.

Yours sincerely,

Haiming Tang

---

## [Decision Letter · Decision Letter 1]

16 Aug 2021

Effect of different long-term fertilizer managements on soil nitrogen fixing bacteria community in a double-cropping rice paddy field of southern China

PONE-D-20-36146R1

Dear Dr. Haiming Tang,

We’re pleased to inform you that your manuscript has been judged scientifically suitable for publication and will be formally accepted for publication once it meets all outstanding technical requirements.

Kind regards,

Balasubramani Ravindran, Ph.D

Academic Editor

PLOS ONE

Additional Editor Comments (optional):

Reviewers' comments:

Reviewer's Responses to Questions

**Comments to the Author**

1. If the authors have adequately addressed your comments raised in a previous round of review and you feel that this manuscript is now acceptable for publication, you may indicate that here to bypass the “Comments to the Author” section, enter your conflict of interest statement in the “Confidential to Editor” section, and submit your "Accept" recommendation.

Reviewer #1: All comments have been addressed

Reviewer #2: All comments have been addressed

2. Is the manuscript technically sound, and do the data support the conclusions?

Reviewer #1: Yes

Reviewer #2: Yes

3. Has the statistical analysis been performed appropriately and rigorously? 

Reviewer #1: Yes

Reviewer #2: Yes

4. Have the authors made all data underlying the findings in their manuscript fully available?

Reviewer #1: Yes

Reviewer #2: Yes

5. Is the manuscript presented in an intelligible fashion and written in standard English?

Reviewer #1: Yes

Reviewer #2: Yes

6. Review Comments to the Author

Reviewer #1: Dear Authors, you have correctly addressed my comments and, therefore, your manuscript can be accepted for publication in my opinion.

Reviewer #2: The authors have adequately addressed my comments and this manuscript is now acceptable for publication

7. PLOS authors have the option to publish the peer review history of their article (what does this mean?). If published, this will include your full peer review and any attached files.

Reviewer #1: **Yes: **Gianluca Caruso

Reviewer #2: No

---

## [Editor Report · Acceptance letter]

23 Aug 2021

PONE-D-20-36146R1 

Effect of different long-term fertilizer managements on soil nitrogen fixing bacteria community in a double-cropping rice paddy field of southern China 

Dear Dr. Tang:

I'm pleased to inform you that your manuscript has been deemed suitable for publication in PLOS ONE. Congratulations! Your manuscript is now with our production department. 

Kind regards, 

on behalf of

Dr. Balasubramani Ravindran 

Academic Editor

PLOS ONE